# An Examination of the Effects of Virtual Reality Training on Spatial Visualization and Transfer of Learning

**DOI:** 10.3390/brainsci13060890

**Published:** 2023-05-31

**Authors:** Kristen Betts, Pratusha Reddy, Tamara Galoyan, Brian Delaney, Donald L. McEachron, Kurtulus Izzetoglu, Patricia A. Shewokis

**Affiliations:** 1School of Education, Drexel University, Philadelphia, PA 19104, USA; tg532@drexel.edu (T.G.); ki25@drexel.edu (K.I.); pas38@drexel.edu (P.A.S.); 2School of Biomedical Engineering, Science and Health Systems, Drexel University, Philadelphia, PA 19104, USA; ylr26@drexel.edu (P.R.); mceachdl@drexel.edu (D.L.M.); 3School of Communication and Journalism, Auburn University, Auburn, AL 36849, USA; bcd0039@auburn.edu; 4College of Nursing & Health Professions, Drexel University, Philadelphia, PA 19104, USA

**Keywords:** spatial navigation, functional brain imaging, fNIRS, mental workload, cognitive load, problem solving, retention, virtual reality

## Abstract

Spatial visualization ability (SVA) has been identified as a potential key factor for academic achievement and student retention in Science, Technology, Engineering, and Mathematics (STEM) in higher education, especially for engineering and related disciplines. Prior studies have shown that training using virtual reality (VR) has the potential to enhance learning through the use of more realistic and/or immersive experiences. The aim of this study was to investigate the effect of VR-based training using spatial visualization tasks on participant performance and mental workload using behavioral (i.e., time spent) and functional near infrared spectroscopy (fNIRS) brain-imaging-technology-derived measures. Data were collected from 10 first-year biomedical engineering students, who engaged with a custom-designed spatial visualization gaming application over a six-week training protocol consisting of tasks and procedures that varied in task load and spatial characteristics. Findings revealed significant small (Cohen’s *d*: 0.10) to large (Cohen’s *d*: 2.40) effects of task load and changes in the spatial characteristics of the task, such as orientation or position changes, on time spent and oxygenated hemoglobin (HbO) measures from all the prefrontal cortex (PFC) areas. Transfer had a large (*d* = 1.37) significant effect on time spent and HbO measures from right anterior medial PFC (AMPFC); while training had a moderate (*d* = 0.48) significant effect on time spent and HbR measures from left AMPFC. The findings from this study have important implications for VR training, research, and instructional design focusing on enhancing the learning, retention, and transfer of spatial skills within and across various VR-based training scenarios.

## 1. Introduction

Engineering is expected to play a pivotal role in the global economy and societal growth over the next decade. Employment opportunities in engineering and engineering-related occupations are expected to increase from 2021 to 2031, resulting in the creation of 91,300 new jobs in the United States over the next decade with approximately 200,900 openings each year coming from growth and replacement needs [1]. It is critical that institutions of higher education be able to prepare adequately educated graduates to fill these positions. Additionally, in The Future of Jobs Report published in October 2020 by the World Economic Forum, several skills were identified as emerging for the coming decade [2]. These skills included problem solving, critical thinking/analysis, and technology use and development, which are also part of the fundamental education and training of engineers [2]. It is imperative that higher education be able to anticipate society’s needs and provide well-educated and skilled graduates in all professions, including engineering.

Spatial visualization abilities (SVA) have been shown to be a predictive factor in Science, Technology, Engineering, and Mathematics (STEM) academic achievement [3,4,5] and retention [5,6]. Spatial visualization is defined by Voyer et al. as “the ability to manipulate complex spatial information when several stages are needed to produce the correct solution” [7] (p. 250). Spatial visualization skills are closely linked to spatial ability—the capability to transform, generate, and recall symbolic information [8]—as well as spatial thinking, which involves the three essential concepts of space, representation tools, and reasoning [9]. In engineering disciplines, spatial abilities contributed to achievement in first-year engineering design beyond other abilities, such as numerical, verbal, and general reasoning [4]. According to Fontaine and De Rosa, students with low spatial visualization skills (SVS) are more likely to drop out of an engineering program [5], highlighting a need to enhance such skills to improve academic performance and retention.

With changing demographics and residual effects from the COVID-19 pandemic, there has been increasing research on developing “new teaching strategies that enhance students’ motivation and commitment and maximize their knowledge acquisition” [10] (p. 2), including virtual reality (VR) training. Digital technologies, such as VR systems, are at the forefront of innovative teaching strategies for supporting SVA skill acquisition in STEM disciplines. According to Oberero-Gaitán et al., the utilization of VR devices in teaching and learning is supported through scientific evidence and enhances educational possibilities in the classroom [11] (p. 3). In support of this assertion, a recent study reported that spatial visualization and mental rotation could be improved through desktop VR training [12]. Thus, a potential use of VR would be to provide training in spatial visualization skills in lieu of the manual manipulation of actual objects in order to promote retention in engineering and other STEM disciplines through enhanced SVS. Recognizing that STEM occupations are projected to grow two times faster than the total of all other occupations between 2021 and 2031, further research is needed on the use of VR to support knowledge acquisition and enhance motivation as well as increase student retention and graduation rates across STEM programs.

The aim of this exploratory study was to examine the effect of VR-based training using spatial visualization tasks on participant performance and mental workload using behavioral (i.e., time spent) and functional near infrared spectroscopy (fNIRS) brain-imaging-technology-derived measures. This study was guided by the following broad research questions:(1)What are the effects of VR training with spatial visualization tasks on participant behavioral performance and learning (i.e., acquisition and transfer)?(2)What are the effects of VR training with spatial visualization on participant mental workload using selected cortical hemodynamic measures (e.g., HbO and HbR) within the prefrontal cortex (PFC) area?

This exploratory study was conducted to advance the understanding of how biomedical engineering students learn SVA and to develop technology-rich learning environments with a pedagogy that fosters learning transfer and student retention.

## 2. Literature Review

### 2.1. Spatial Visualization

Various studies have reported that engaging students in authentic spatial visualization projects can enhance deeper learning [13,14,15]. Deeper learning has been defined in terms of key competencies, including the ability to think critically and solve complex problems [16,17,18]. Such learning is crucial to transferring skills both within and beyond academic settings. It is therefore essential to design and implement innovative curricula that will effectively foster transferable spatial visualization skills, particularly during the first year of college. Spatial visualization tutorials and workshops have been shown to generate overall improvements in spatial ability for first-year engineering students [13,14,19]. Furthermore, studies have reported that those students who can increase their spatial visualization skills demonstrate greater self-efficacy, which, in turn, appears to positively affect retention in engineering [20,21,22]. Retention is a metric that reflects a student’s ability to continue from one term to the next through to graduation. According to Gómez-Tone et al. [14], “Strong spatial skills have been shown to be associated with overall success and achievement, but also retention in engineering programs” (p. 4).

### 2.2. Purdue Spatial Visualization Test: Rotations (PSVT:R)

The Purdue Spatial Visualization Test: Rotations (PSVT:R), which involves rotating objects, was selected for this study since it is one of the most widely used spatial visualization tests in the engineering education research community [20,23]. The PSVT:R was designed by Guay in 1976 to determine a “student’s ability to visualize and recognize orthographic drawings” [24] (p. 161). The PSVT:R assessment includes 30 multiple-choice questions in which students mentally rotate 3D geometric shapes through specified rotations (see Figure 1 for an example).

### 2.3. Virtual Reality

Providing learners with alternatives to traditional lectures is becoming increasingly important in 21st century classrooms that are characterized as being more individual-oriented, task-based, and technology-enhanced environments [25]. Two of the Grand Challenges proposed by the National Academy of Engineering in 2017 were Enhance Virtual Reality and Advance Personalized Learning (NAE Grand Challenges for Engineering, 2017). Advances in virtual reality (VR) technologies can address both challenges, allowing the creation of innovative and interactive personal learning environments that can simulate real environments in terms of sight, hearing, and touch [26]. VR environments provide novel approaches to educational content delivery and learner engagement. Studies have shown that interactive media tools can have a positive effect on the understanding of the learning content, learner motivation, engagement, and creative self-expression [27,28,29]. In engineering education, the application of head-mounted displays for immersive VR experiences has led to learning enhancements due to the increased interactivity of simulation tasks [25,30]. Studies have also indicated that teaching via interactive 3D models in VR environments have advantages in terms of efficient learning, transfer, higher motivation, and greater engagement when compared to the traditional approaches [26,28,31]. A recent study by Kuznetcova and colleagues [32] showed that VR intervention improved middle school students’ visuospatial self-efficacy. Another related study involving three comparison groups (i.e., VR, Desktop VR, and 2D) showed increased usability and learning outcomes for the immersive VR group as compared to the other groups [33]. Virtual reality training has also been used to promote spatial ability in older adults with mild cognitive impairment [34], allocentric space representations in young males [35], as well as spatial visualization and mental rotation in a controlled experiment [12]. The latter study used college-age students (27 females and 22 males, ages 19–28). The evidence clearly supports the use of VR training in an effort to enhance spatial visualization ability in first year college students.

The efficacy of VR in enhancing learning can be evaluated by assessing the effects of task and environmental characteristics on performance and mental workload [28]. Performance is commonly evaluated using behavioral measures, such as time, speed, and accuracy. A comparison of VR training versus traditional training has demonstrated a faster improvement in performance during skill acquisition, and greater skill transfer when using VR [36]. Several methods currently exist to measure mental workload. The oldest and the most widely used approaches have involved subjective ratings. These methods are effective but are usually performed after the completion of a task and require participant introspection. Alternatively, physiological (i.e., heart rate, eye tracking, etc.) and neurophysiological measures (i.e., electroencephalography, functional near infrared spectroscopy) enable the assessment of mental workload during task execution.

### 2.4. Functional near Infrared Spectroscopy (fNIRS)

Functional near infrared spectroscopy (fNIRS) is a non-invasive portable brain imaging technology that measures hemodynamic changes within the cerebral cortex in response to sensory, motor, and cognitive stimuli [37,38,39]. Previous studies have associated fNIRS measures with varying workload levels during cognitive tasks [39,40,41,42,43,44]. Furthermore, studies with fNIRS have shown a significant reduction in the activation within the PFC with increasing familiarity with the task at hand and improved performance, thus providing a quantitative measurement of trainees’ expertise development [45]. Research has shown that mental workload is sensitive to task-related features such as difficulty level, the order of the tasks (e.g., blocked vs. random practice), and the type of task (e.g., learning vs. transfer task) [39,43,44].

Based on the studies shared in the literature, we decided to implement the Purdue Assessment in VR for a hands-on application of special visualization practice and selected fNIRS as the neurophysiological biometric to assess participants’ mental workload during the VR experience.

## 3. Materials and Methods

### 3.1. Participants

Ten participants between the ages of 18 and 20 consented to participate in a 6-week immersive virtual reality (VR) intervention study. The study was approved by the Institutional Review Board (IRB) of Drexel University. Out of the 10 participants, 7 were male. Recruited individuals were right-handed; had either normal or corrected-to-normal vision; had no history of seizures, head injury, or neurological dysfunction; had no history of depression, schizophrenia, or social phobia; had not had previously admitted to an alcohol/drug treatment program or diagnosed as suffering from of alcohol/drug abuse; were not taking medications that could affect or alter brain activity (i.e., sleeping pills, Valium, or Xanax); and were not pregnant.

### 3.2. VR Protocol

The 6-week VR protocol is shown in Figure 2 broken down by week. A total of three different protocols were implemented. Each block, regardless of color, consisted of 12 target shapes, 3 extra shapes, and 12 panel boxes, where the subject was required to match the correct shape to the correct box (see Figure 3). The first protocol, as indicated by orange blocks, consisted of three different tasks with increasing difficulty. The level of difficulty was manipulated by making the shapes and panel boxes more complex. The Purdue Spatial Visualization Task identifies levels of increasing difficulty (complexity). Thus, we based our protocol and levels of difficulty on the Purdue criteria [30]. The second protocol, as indicated by blue blocks, consisted of three spatial manipulations of either the target shapes, the panel boxes, or both. The first and fourth blue blocks, noted as control in Figure 2, consisted of a new set of target shapes and panel boxes. The second blue block, noted as position, consisted of the same setup of target shapes as the first block with changes in the position of the panel boxes. The third blue block, noted as orientation, consisted of the same setup of panel boxes as the second block with changes in the orientation of the target shapes. The fourth and sixth blue block, noted as position + orientation, consisted of changes in the position of the panel box and the orientation of the target shapes. The third protocol, as indicated by green blocks, consisted of three control blocks and three blocks where the position of the panel box and the orientation of the target shapes from the preceding control blocks were manipulated. The primary difference between the second protocol and third protocol is that a new set of target shapes and panel boxes was used. The final protocol implemented during week 6 utilized a randomly selected block from any of the three protocols administered during weeks 1 through 5.

### 3.3. VR and fNIRS Equipment

For hardware, this study utilized an Oculus Rift with the Touch Reality System and a Razer Blade gaming laptop. The continuous-wave fNIRS device (fNIRS Imager 1200; fNIR Devices LLC, Potomac, MD, USA) utilized in this study emitted light at peak wavelengths of 730 and 850 nm, sampled every 2 Hz, and consisted of 4 light-emitting diodes (LEDs) and 10 photodetectors. The detector and light source combination results in a total of 16 cerebral measurement locations and enabled monitoring of hemodynamic changes in the PFC region (see Figure 4).

### 3.4. fNIRS Signal Processing and Feature Extraction

fNIRS signals are affected by noise arising from motion, ambient light, saturation, and physiological factors, such as cardiac activity, respiratory function, and Mayer waves (low-frequency spontaneous oscillations in arterial blood pressure) [47]. To remove the effects due to motion, ambient light, and saturation, channels were rejected if differences between light intensity measurements from the two wavelengths were less than 70, average light intensity measurements were less than 700, and/or the standard deviations (SDs) of light intensity measurements were greater than 3 SD [48,49]. To remove effects due to physiological sources, a low-pass filter with a cut-off frequency of 0.09 Hz was employed [47]. A modified Beer–Lambert law and local baselining were then used to calculate relative concertation changes in HbO and HbR [50]. Average HbO and HbR measures per block (protocol X level) were extracted from Left Dorsolateral PFC (LDLPFC—channels 1 to 4), Left Anterior Medial PFC (LAMPFC—channels 5 to 8), Right Anterior Medial PFC (RAMPFC—channels 9 to 12), and Right Dorsolateral PFC (RDLPFC—channels 13 to 16). The location of these regions over the PFC is shown in Figure 4.

### 3.5. Statistics

A longitudinal, repeated-measures research design was used which included a hierarchical nesting structure. In addition, missing data were assessed for characterization and percent missed (determined to be missing at random as shown in Table A1). Given that data are missing at random, linear mixed-effects models provide a viable statistical technique to analyze the data without imputations [51]. Therefore, linear mixed-effects regression (LMER) modelling was used [52], with nlme and emmean*s* used to perform LMER analyses [51,53]. Dependent variables evaluated consisted of one behavioral (time spent—s) and eight fNIRS measures (HbO and HbR—uM measures from LDLPFC, LAMPFC, RAMPFC, and RDLPFC). The dependent variable was normalized (i.e., centered and scaled) prior to statistical evaluation. The data were examined in four separate analyses to assess task load, spatial characteristics, transfer, and training, respectively. For the first analysis, data associated with protocol 1 from weeks 1 to 5 (see blocks highlighted by a black bar associated with task load in Figure 2) were extracted to investigate the effect of changes in task load (easy, medium, hard) on dependent variables (see Model 1). Post hoc analysis for this investigation consisted of three task load comparisons (easy vs. medium, medium vs. hard, and easy vs. hard). For the second analysis, data associated with the first four blocks of protocol 2 from weeks 3 to 5 (see blocks highlighted by a black bar associated with spatial characteristics in Figure 2) were extracted to investigate the effect of changes in spatial characteristics (control, position change, orientation change, and position + orientation change) on dependent variables (see Model 2). Post hoc analysis consisted of six comparisons (control vs. position, control vs. orientation, control vs. position + orientation, position vs. orientation, position vs. position + orientation, and origin vs. position + orientation).

For the third analysis, data associated with last two blocks of protocol 2 and all blocks of protocol 3 from week 4 (see blocks highlighted by a black bar associated with transfer in Figure 2) were extracted to investigate the effect of introducing new scenarios with same task objective (see Model 3). Post hoc analysis consisted of two within comparisons (Old: Control vs. Position + Orientation; New: Control vs. Position + Orientation) and two across comparisons (Control: Old vs. New; Position + Orientation: Old vs. New). Finally, for the fourth analysis, data from weeks 1 through 5 were labelled as acquisition, while the data from week 6 were labelled retention (see blocks highlighted by a black bar associated with training in Figure 2) to investigate training effect (see Model 4). Post hoc analysis consisted of three total comparisons, with one comparison (Acquisition vs. Retention) per protocol:

The LMER analyses were based on the following general statistical models and equations.

**Model 1:** Random intercept, fixed effect: task load (easy, medium, hard).

The random intercept model is the simplest model in which the *i*th observation of the DV in the *j*^th^ task load and the *k*^th^ subject is depicted as:DV*_ijk_* = 1 + Task Load*_j_* + (1|Subject*_k_*) + ε*_ijk_*(1)
where “1” is the intercept representing the overall mean, “Task Load” is a fixed effect, and “Subject” indicates a random effect due to the subject that the *i*th observation was in.

**Model 2:** Random intercept, fixed effect: spatial characteristics (control, origin, position, position + origin).
DV*_ijk_* = 1 + Spatial Characteristics*_j_* + (1|Subject*_k_*) + ε*_ijk_*(2)
Model 2 is a similar model to Model 1 with a change in the fixed effect to “Spatial Characteristics”.

**Model 3:** Random intercept, fixed effects: protocol (old: week 4—protocol 2; new: week 4—protocol 3); protocol: dependency (old: control, position + origin, new: control, position + origin); protocol: dependency interaction.
DV*_ijk_* = 1 + Protocol*_j_* + Protocol: Dependency*_jl_* + (1|Subject*_k_*) + ε*_ijlk_*(3)
where “1” is the intercept representing the overall mean; “Protocol” is a fixed effect; protocol: dependency is an interaction fixed effect; and “Subject” indicates a random effect due to the subject that the *i*th observation was in.

**Model 4:** Random intercept, fixed effect: session (acquisition: week 1—5, retention: week 6), session: protocol (acquisition: protocol 1, protocol 2, protocol 3, retention: protocol 1, protocol 2, protocol 3) interaction.
DV*_ijk_* = 1 + Session*_j_* + Session: Protocol*_jl_* + (1|Subject*_k_*) + ε*_ijlk_*(4)
Model 4 is a similar model to Model 3 with a change in the fixed effects to “Session” main effect and “Session: Protocol” interaction fixed effect.

The heterogeneous AR1 covariance model is a first-order autoregressive structure with heterogeneous variances. Defining the correlations between any two elements results in a correlation coefficient *r*, while correlations between two elements that are separated by a third element (time point) will be (*r*2) with this pattern continuing. Within our study, we assumed that the adjacent observations (time points) on the same subject (performer) will have errors with a higher correlation than observations that are not adjacent or further apart. Additionally, since the covariance structures assume different variances for each time point, these different variances are labeled “heterogeneous” covariance structures. In the nlme R package, we chose the coAR1 function because the function uses a discrete-time first-order autocorrelation model [51].

Due to strong correlations observed between measures that varied across weeks, all models used a heterogenous AR1 covariance structure [51]. Tests of the assumptions of homogeneity of variance and normality of residuals along with random effects were conducted using visual inspection. If model predictions showed heteroscedasticity or a non-normal distribution, then log10 transformations were performed on the response variables. The significance of fixed effect terms was evaluated using likelihood ratio tests, where the full effects model was compared against a model without the effect in question. For example, while investigating the effect of task load, a reduced (null–random intercept) model of 1 + (1|Subject) was compared against the (random intercept, fixed effect model) 1 + Task Load + (1|Subject). Maximum likelihood estimation was used to conduct likelihood ratio tests, while restricted maximum likelihood was used to evaluate post hoc comparisons. For all statistical analyses, the level of significance was set at α = 0.05. Adjustments using false discovery rate (FDR) were made on *p*-values to account for Type I error inflation per dependent variable. Cohen’s *d* was used to examine post hoc effects [54]. A *d* of 0.2 is considered a small effect, while 0.5 and 0.8 represent medium and large effects, respectively.

## 4. Results

### 4.1. Effect of Task Load on Behavioral and fNIRS Measures

Task load had a significant effect on time spent ($\chi^{2}\left(2\right)$ = 191.50, *p* < 0.001), with increases observed from easy to medium, medium to hard, and easy to hard (see leftmost plot in Figure 5 and Table 1). Task load had a significant effect on all PFC areas for HbO measures (LDLPFC: $\chi^{2}\left(2\right)$ = 33.56, *p* < 0.001; LAMPFC: $\chi^{2}\left(2\right)$ = 28.55, *p* < 0.001; RAMPFC: $\chi^{2}\left(2\right)$ = 35.01, *p* < 0.001; RDLPFC: $\chi^{2}\left(2\right)$ = 40.76, *p* < 0.001), and only the LDLPFC ($\chi^{2}\left(2\right)$ = 6.67, *p* = 0.036), RAMPFC ($\chi^{2}\left(2\right)$ = 6.07, *p* = 0.048), and RDLPFC ($\chi^{2}\left(2\right)$ = 10.24, *p* = 0.006) areas for HbR measures. Post hoc comparisons, as shown in Table 1 and Figure 5 and Figure 6 (topographical maps), revealed significant increases from (i) easy to medium conditions in only RAMPFC and RDLPFC; (ii) medium to hard in all regions; and (iii) easy to hard in all regions. Although significant, most detected effects ranged from small (*d* ~ 0.1) to small–moderate (*d* = 0.31 to 0.51).

### 4.2. Effect of Spatial Characteristics on Behavioral and fNIRS Measures

Changes in spatial characteristics had a significant effect on time spent ($\chi^{2}\left(3\right)$ = 35.68, *p* < 0.001) (see first plot of top panel in Figure 7). Post hoc comparisons, as shown in Table A2, revealed that the time spent by the control group significantly decreased during position change, orientation change, and during position + orientation change. Significant increases in time spent were observed from position change or orientation change to position + orientation change. No differences were observed between position and orientation changes. Spatial characteristics had significant differences with effect sizes ranging from small to large (*d* = 0.24 to *d* = −0.81), respectively.

Spatial characteristics had a significant effect on all PFC areas of HbO measures (LDLPFC: $\chi^{2}\left(3\right)$ = 21.24, *p* < 0.001; LAMPFC: $\chi^{2}\left(3\right)$ = 26.04, *p* < 0.001; RAMPFC: $\chi^{2}\left(3\right)$ = 39.88, *p* < 0.001; RDLPFC: $\chi^{2}\left(3\right)$ = 35.87, *p* < 0.001), and only the RDLPFC area ($\chi^{2}\left(3\right)$ = 8.15, *p* = 0.043) of HbR measures. Post hoc comparisons, as shown in Table A2 and the bottom panel of Figure 7, revealed significant increases from (i) control to position change in LDLPFC and RDLPFC; (ii) control to orientation change in all areas; (iii) control to position + orientation change in all areas; (iv) position to position + orientation change in all areas; (v) orientation to position + orientation change in all areas; and (vi) position to orientation change in only RDLPFC. Effect sizes ranged from small–moderate (d = −0.34) to large (−1.09). Table A2 provides descriptive statistics associated with each change in spatial characteristics and the differences between them.

### 4.3. Effect of Transfer on Behavioral and fNIRS Measures

The introduction to a new scenario with same the task objective as previously assessed scenarios resulted in a significant interaction effect between the protocol and dependency on time spent ($\chi^{2}\left(2\right)$ = 18.52, *p* < 0.001) (see first plot of Figure 8). Post hoc comparisons revealed significant increases from control to position + origin change in the new scenario (adj. *p* < 0.001, *d* = 0.83), and increases from the old to new scenario for the control dependency (adj. *p* < 0.001, *d* = −1.37). Although the interaction effect between the protocol and dependency was not significant for either HbO or HbR for any PFC areas, the main effect of the protocol was significant for RAMPFC ($\chi^{2}\left(1\right)$ = 4.55, *p* = 0.033) for HbO measures, specifically with decreases observed during performance of the new scenario (see fourth pair of plots of Figure 8).

### 4.4. Effect of Training on Behavioral and fNIRS Measures

As shown in Figure 9, changes in performance and brain activity were observed across weeks as per the protocol. However, statistical evaluation revealed significant interactions between session and protocol only on time spent ($\chi^{2}\left(4\right)$ = 78.99, *p* < 0.001) and HbR measures from LAMPFC ($\chi^{2}\left(4\right)$ = 10.87, *p* = 0.028). Post hoc comparisons between sessions per protocol for time spent revealed significant decreases in time spent during the retention session for protocols 1 (adj. *p* = 0.009, *d* = 0.58), 2 (adj. *p* = 0.025, *d* = 0.48), and 3 (adj. *p* = 0.009, *d* = 0.64), yielding approximately moderate effect sizes across each protocol. Unlike the trends observed in behavioral measures, no post hoc comparisons were significant for HbR measures from LAMPFC.

## 5. Discussion

The purpose of this exploratory study was to examine the effect of customized immersive VR spatial visualization training on participant acquisition, transfer, and mental workload using selected performance and cortical hemodynamic measures. Standard behavioral performance metrics (total time to task completion (s)) were obtained from a customized VR task based on the Purdue Spatial Visualization Test: Rotations (PSVT:R), while cortical hemodynamic responses were measured by a wearable optical brain imaging device that was worn on the participants’ foreheads for all trials. The PSVT:R test assesses spatial visualization ability (SVA), an important skill for students enrolled in engineering and related disciplines that impacts those students’ success and retention in STEM-related disciplines. The effects of the customized VR on task load, spatial characteristics, transfer, and training with a focus on the influence of cortical hemodynamics on SVA are discussed below.

A primary feature of skill acquisition is the impact of task load or task difficulty on goal attainment and prefrontal cortex activation [42,43,47,57]. Behaviorally, we found that time increased monotonically with increased task load (difficulty) for the easy, medium, and hard tasks as noted in Table 1 and Figure 5. The effect sizes (see Table 1) ranged from large to very large (Cohen’s *d*: −0.78 to −2.61) for the easy to hard tasks, respectively. Our task load findings on time were similar to those detected by Haji et al. with simple and complex lumbar puncture surgical simulated tasks [47], insofar as increases in task difficulty result in additional time to complete the task. The effect of task load on the fNIRS biomarkers of HbO and HbR are depicted in Table 1 and Figure 5. As reported, all four regions of the prefrontal cortex—LDLPFC, LAMPFC, RAMPFC, and RDLPC—were significantly impacted by the task load manipulation (*p* < 0.001), where there were small to moderate effects (*d* = −0.23 to −0.51) for the contrasts of the medium to hard and hard to easy tasks. The right prefrontal cortex (RDLPFC and RAMPFC) showed increases in activation for HbO for the easy to medium task load conditions. In terms of HbR, task load resulted in differences for the LDLPFC, RDLPFC, and RAMPFC, albeit with a smaller impact.

To understand the association between task load and brain activity, it is critical to be familiar with the fundamental neurovascular coupling physiological principles. These principles describe the relationship between neuronal activity and cerebral hemodynamics for fNIRS measurements, allowing insight into the role of brain activation for the interpretation of activities and tasks with varying difficulty levels or task loads [57,58,59]. The physiological foundation of neurovascular coupling links oxygen requirements and the delivery of oxygen used for glucose metabolism through the neuronal activation process. This increased neuronal activation process results in an increase in cerebral blood flow that carries oxygen to the needed region through the biomarker oxygenated hemoglobin (HbO). The conversion process of HbO to deoxygenated hemoglobin (HbR) then yields the release of oxygen. Since HbO and HbR are the primary absorbers of the near-infrared light bands in fNIRS cognitive functioning implementation studies, they are used to assess the relative concentrations of biomarkers, allowing a calculation of brain activity to be correlated with task performances.

In his review of the role of neuroimaging for cognitive load theory, Whelan summarized various fMRI papers indicating that for intrinsic cognitive load, the dorsolateral prefrontal cortices (right and left) were the primary regions of activation associated with attentional control and working memory [59] that were critically modulated by cognitive task load and/or task difficulty [56]. Figure 6 denotes the functions of the relevant regions of the prefrontal cortex that have been associated with the different types of cognitive load. Specifically, the dorsolateral prefrontal cortex (DLPFC) and ventromedial PFC are the regions associated with the task load manipulations in this VR study (specific functions of the regions are specified in Figure 4—bottom panel). Importantly, the DLPFC is thought of as the cognitive load control mechanism where the DLPFC structure is considered the manager of the buffering, retrieval, and computation of task-related information [49]. The intrinsic load is associated with the performer and task as well as the interaction between them. The functions of the right DLPFC and the left DLPFC (identified in Figure 4—bottom panel) facilitate the processing of task-relevant information and integrate the processes of attention, short-term memory, retrieval, and decision making [42,55,56,60]. When Zhu and colleagues assessed the combined impact of galvanic skin response, eye-movement tracking, and EEG with varying arithmetic tasks designed using the NIH cognitive toolbox [60,61,62]; they found that frontal brain activation increased monotonically although with varying slopes as the task difficulty or load increased [60]. Our task load findings showed a stronger slope of changes from our medium to hard task while the easy to medium task had either no change or a slight reduction in slope (see Figure 5). Our findings support Zhu’s suggestion that it is important to discern differences in brain activity patterns when assessing the decoding of cognitive load, given that brain activity patterns manifest in various ways to different types of cognitive tasks and levels of task difficulty or load [60,61,62].

Four manipulations of the spatial characteristics of the immersive virtual reality tasks were assessed in our experiment: (1) control; (2) position change; (3) orientation change; and (4) position + orientation change. Spatial characteristics yielded important differences in time spent for task completion as depicted in Table A2 and Figure 7. As expected, behavioral data showed that time spent in the control condition decreased during the position and during the orientation change. In addition, when the transition was from control to position and orientation change, there was a decrease in time spent on the task. The spatial characteristics and task manipulations were particularly important in our study given that the tasks were designed based on the PSVT:R for the skill assessment of students enrolled in engineering and STEM-related areas in terms of spatial visualization abilities. The task spatial characteristics significantly impacted all PFC regions for HbO while they reliably influenced only the RDLPFC for HbR.

Learning is considered to be robustly determined through memory and generalizability assessments [39,43]. In the current investigation, we introduced a new scenario with the same task objective as the previously assessed scenarios; namely, the assessment of the data associated with the last two blocks of the week 2 protocol and all blocks of the week 3 protocol that were tested in week 4 as depicted in Figure 2. The transfer assessment yielded a significant interaction of protocol X condition on the behavioral measure of time spent on the task. There were large effects of a reduction in time spent on task that were reliably detected from control to position and orientation changes in the new scenario, which may be considered an interpolation of the spatial characteristics that were acquired during the week 2 protocol. An interpolation transfer of this type is comparable to the transfer of behavioral scores in a VR surgical coordination task in Shewokis et al. [42,43], as well as the findings of Galoyan et al. [39] of reduced total time in week 2 relative to week 1 in a virtual spatial navigation transfer game.

In terms of brain imaging and decoding of cognitive load [43,56,62], we posit that transfer is associated with germane cognitive load [55], which is realized by an activation of the ventromedial prefrontal cortex (see Figure 4 bottom panel and Figure 6). The ventromedial prefrontal cortex incorporates the integration of information and allows for multiple items in working memory, which are functions necessary for the facilitation of generalizability or transfer. Although we did not detect any significant hemodynamic differences across the VR spatial characterization tasks (see Figure 8—third (HbO-LAMPFC) and fourth (HbO-RAMPFC) panels), there was a trend for transfer from the old to new protocols with the RAMPFC with reduced HbO activation. The lack of differences may be attributed to the high variability in hemodynamic responses. Future work with increased training on the tasks, perhaps to a criterion level of behavioral performance, may elicit stronger hemodynamic responses that are more stable.

Like traditional training findings, studies have demonstrated positive connections between task load and mental workload in VR environments [63]. On the other hand, studies have reported contradictory findings when comparing mental workload in VR environments versus 2D environments, with some reporting greater mental workload in VR environments [63,64] and others reporting no differences [65,66]. Similarly, studies have reported contradictory findings when comparing mental workload during VR training with traditional training. Some studies reported decreases in mental workload and better learning because of VR training [67], while others reported increases in mental workload and poorer learning [63].

From the inception of our study through its culmination, we were interested in determining if the training protocols allowed for improved performance and learning with this applied task performed in a virtual environment. We found that there was a significant interaction between session and protocol for the behavioral measure of time spent on the task, yielding moderate effects with a time decrease for protocol 1. In addition, there was a significant interaction effect of session and protocol for HbR in the LAMPFC, indicating the activation of the neurovascular coupling process and the release of oxygen. However, as depicted in Figure 9—last series of panels on HbO—the high variability in responses to the spatial characteristics of the virtual tasks may be due to having insufficient practice trials (6 weeks including testing and orientation to the tasks) along with a small size (N = 10).

There were two primary limitations to this study. First, the small sample size and sampling method imply that our findings cannot be generalized to broader populations. Instead, our results will be used to inform an expanded next phase of this research inquiry. Second, the immersive virtual reality intervention was produced on a limited budget. With additional resources, a more graphically robust intervention could further enhance the nature of how learners interact with spatial visualization objects.

## 6. Conclusions

In engineering and related STEM disciplines, spatial visualization skills have been shown to be a critical factor in enhancing learner academic achievement and retention [3,4,5,6]. Findings from this study revealed important small (Cohen’s *d*: 0.10 to large 2.40) effects of task load and changes in the spatial characteristics of the task, such as orientation or position changes, on time spent and HbO measures from all the prefrontal cortex (PFC) areas when students underwent varying levels of spatial visualization training using VR simulations of object manipulation based upon the Purdue Spatial Visualization Test. Transfer had a large (*d* = 1.37) effect on time spent and HbO measures from right anterior medial PFC (AMPFC); while training had a moderate (*d* = 0.48) significant effect on time spent and HbR measures from left AMPFC. High variability in some measurements could be due to limitations in the number of practice trials combined with the small sample size. In order to come to more definitive conclusions, further research will be needed to examine the specific effects of different task characteristics on participant mental workload and behavioral performance during VR training. Future research will also be necessary to explore the extent to which learners generalized the acquired skills from VR to other non-VR learning scenarios.

## Figures and Tables

**Figure 1 brainsci-13-00890-f001:**
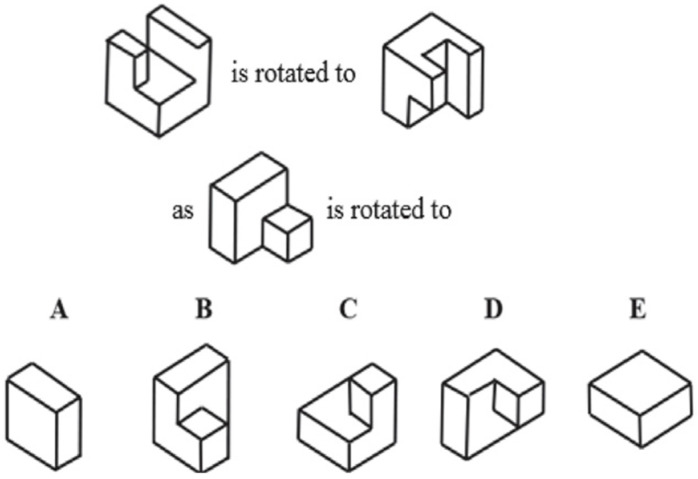
Example of a PSVT question.

**Figure 2 brainsci-13-00890-f002:**
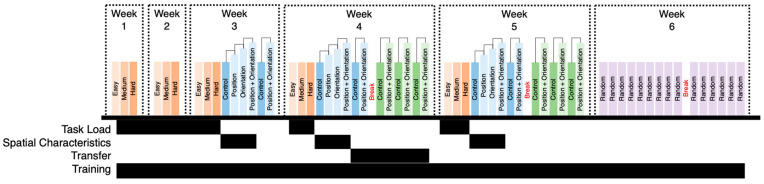
Six-week experimental protocol consisting of three different task sets (orange, blue, and green blocks). The first set (orange blocks) consisted of changes in task load. The second and third sets (represented via blue and green blocks) consisted of changes in spatial characteristics. Purple blocks (week 6) represent tasks sampled from either of the task sets. The black bars under each of the colored blocks represent which data were used in linear mixed-effects models to investigate the effect of task difficulty, spatial characteristics, transfer, and training.

**Figure 3 brainsci-13-00890-f003:**
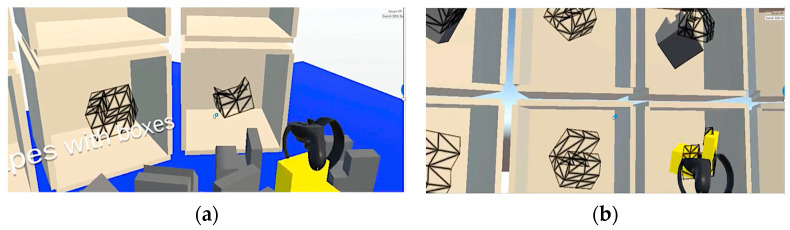
VR application: picking up Purdue cube (**a**) and placing cube in selection box (**b**).

**Figure 4 brainsci-13-00890-f004:**
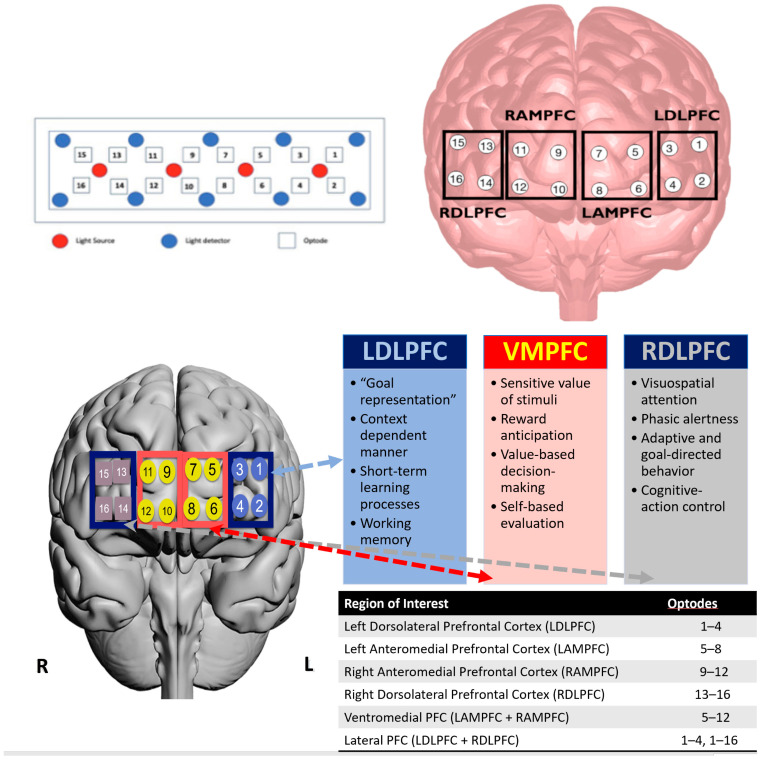
fNIRS sensor layout and the grouping of fNIRS channels based on functional regions—Right Dorsolateral Prefrontal Cortex (RDLPFC), Right Anterior Medial Prefrontal Cortex (RAMPFC), Left Anterior Medial Prefrontal Cortex (LAMPFC), and Left Dorsolateral Prefrontal cortex (LDLPFC) (**top**); topographical overlay across functional regions of the prefrontal cortex with associated functions and sensor groupings (**bottom**). Adapted graphic from Getchell and Shewokis [46].

**Figure 5 brainsci-13-00890-f005:**
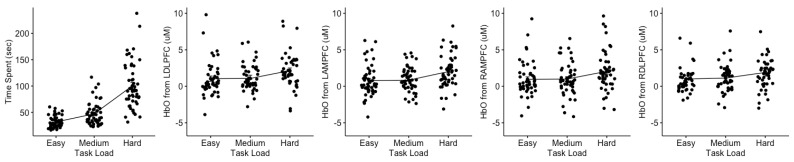
Effect of changes in task load on time spent and HbO measures from Left Dorsolateral Prefrontal Cortex (LDLPFC), Left Anterior Medial Prefrontal Cortex (LAMPFC), Right Anterior Medial Prefrontal Cortex (RAMPFC), and Right Dorsolateral Prefrontal Cortex (RDLPFC). Data represented using mean and standard error of mean. Data were pooled across weeks 1 through 5 from protocol 1.

**Figure 6 brainsci-13-00890-f006:**
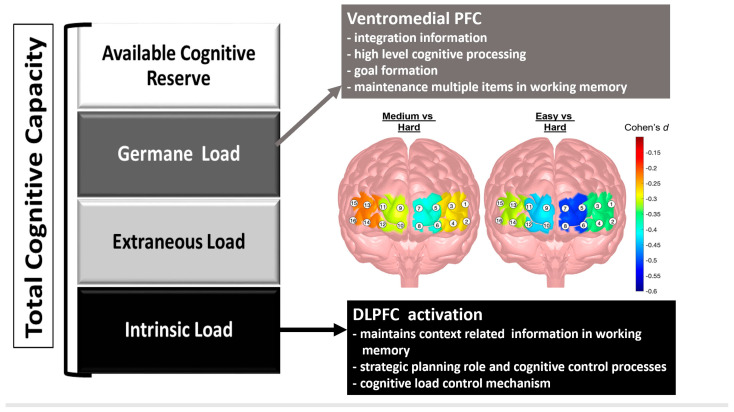
Schematic of Total Cognitive Capacity with Suggested Activation Functions of the Dorsolateral Prefrontal Cortex (DLPFC), Ventromedial Prefrontal Cortex, Associated Types of Cognitive Load based on Cognitive Load Theory, and Contrast Topographical Maps of the Cohen’s *d* Effect Sizes Comparing Significant Task Load Comparisons [43,55,56]. Adapted graphic [46].

**Figure 7 brainsci-13-00890-f007:**
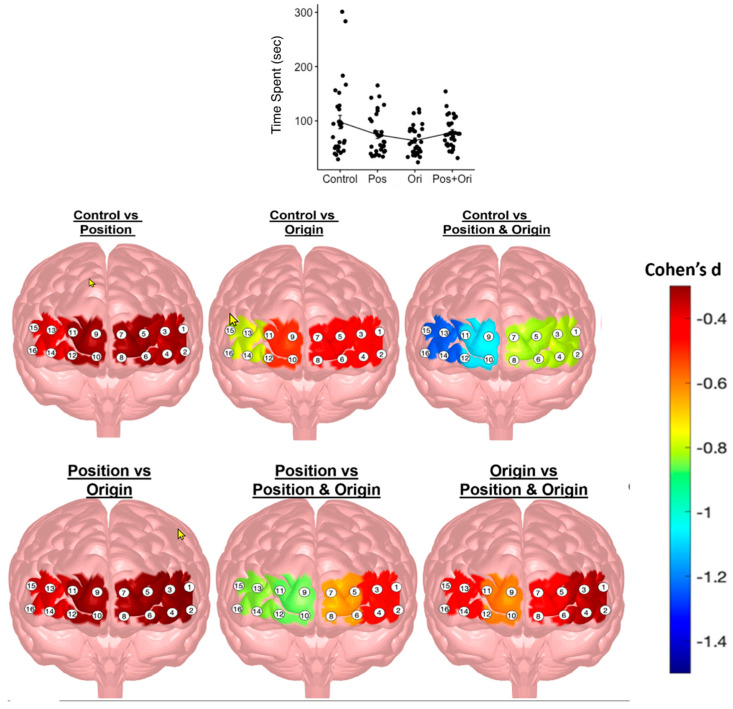
Effect of changes in spatial characteristics on time spent and HbO measures from LDLPFC, LAMPFC, RAMPFC, and RDLPFC. Data were pooled across weeks 3 through 5 from first four blocks of protocol 2 and represented using mean and standard error of mean (top panel). Topographical maps of contrast effect sizes (Cohen’s *d*) for spatial characteristics of HbO for all regions of interest in the PFC (bottom panel).

**Figure 8 brainsci-13-00890-f008:**
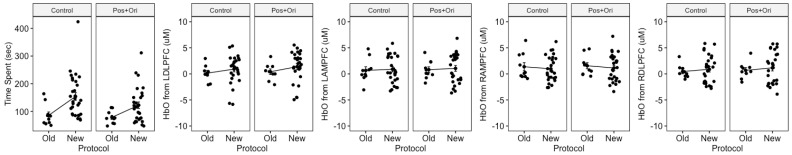
Effect of introduction to new scenarios on time spent and HbO measures from LDLPFC, LAMPFC, RAMPFC, and RDLPFC. Data plotted for old are from last two blocks of protocol 2, and new are from blocks of protocol 3 during week 4, and represented using mean and standard error of mean.

**Figure 9 brainsci-13-00890-f009:**
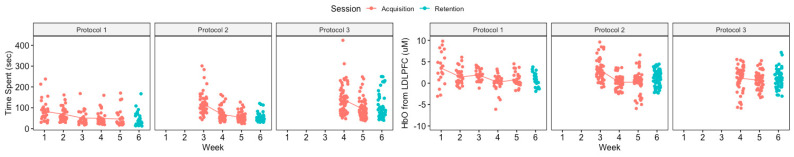
Effect of training on time spent and HbO measures from LDLPFC. Data are represented using mean and standard error of mean.

**Table 1 brainsci-13-00890-t001:** Descriptive statistics associated with each task load level and differences between them.

		Task Load	Mean	SD	Easy–Medium		Medium–Hard		Easy–Hard	
					adj. *p*	*d*	adj. *p*	*d*	adj. *p*	*d*
**Behavioral**	**Time Spent (s)**	**Easy**	**31.72**	11.24	<0.001	−0.78	<0.001	−1.63	<0.001	−2.40
		Medium	47.30	21.78						
		Hard	100.27	44.00						
HbO	LDLPFC (uM)	Easy	1.04	2.39	0.140	−0.08	<0.001	-0.27	<0.001	−0.35
		Medium	1.08	2.12						
		Hard	2.15	2.43						
	LAMPFC (uM)	Easy	0.77	2.14	0.160	−0.12	<0.001	−0.39	<0.001	−0.51
		Medium	0.84	2.02						
		Hard	2.02	2.28						
	RAMPFC (uM)	Easy	0.96	2.33	0.044	−0.13	<0.001	−0.31	<0.001	−0.44
		Medium	1.00	2.22						
		Hard	2.05	2.69						
	RDLPFC (uM)	Easy	1.00	2.19	0.036	−0.09	<0.001	−0.23	<0.001	−0.31
		Medium	1.13	1.90						
		Hard	1.89	2.39						
HbR	LDLPFC (uM)	Easy	0.20	1.00	0.211	−0.07	0.028	0.15	0.211	0.08
		Medium	0.39	1.22						
		Hard	0.08	1.27						
	LAMPFC (uM)	Easy	0.00	0.98	0.841	−0.02	0.493	0.09	0.493	0.08
		Medium	0.18	1.24						
		Hard	−0.04	1.27						
	RAMPFC (uM)	Easy	0.04	0.72	0.950	0.01	0.045	0.20	0.045	0.21
		Medium	0.15	1.02						
		Hard	−0.06	1.07						
	RDLPFC (uM)	Easy	−0.12	1.07	0.871	0.01	0.007	0.14	0.007	0.14
		Medium	0.07	1.49						
		Hard	−0.24	1.63						

## Data Availability

The data presented in this study are available on request from the corresponding author. The data are not publicly available due to privacy concerns regarding student information on enrollment and retention.

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
