# Peer review of "An Examination of the Effects of Virtual Reality Training on Spatial Visualization and Transfer of Learning"

_brainsci, 2023, doi:10.3390/brainsci13060890_

Round 1

Reviewer 1 Report (Previous Reviewer 1)

The authors have considered my concerns and acceptably revised their manuscript. I recommend publication.

Reviewer 2 Report (Previous Reviewer 2)

The authors have properly addressed all my previous comments. The manuscript is much more clear now and its quality greatly improved.

This manuscript is a resubmission of an earlier submission. The following is a list of the peer review reports and author responses from that submission.

Round 1

Reviewer 1 Report

The authors investigated the effect of virtual reality training on the behavioral and neural functioning of first-year biomedical engineering students. They used a spatial visualization task and measured the time spent on the task as well as hemodynamic changes in frontal brain regions using functional near infrared spectroscopy (fNIRS). Students were trained over 6 weeks with a task that varied in task workload and spatial characteristics. Behavioral findings revealed an increase in time spent with increasing task load as well as hemodynamic changes in all prefrontal brain regions.

The study is interesting and the paper is generally very well written. However, I have a few concerns.

1.       The authors should add a conclusion to their abstract and not only describe the main findings.

2.       The introduction describes the task and the procedure as well as the NIRS measurement in detail. However, a description of the detailed research question(s) and the hypotheses is missing at the end of the introduction. Also, a clear rationale for using the NIRS method should be added. If the study is mainly exploratory, the authors should explicitly state this in their introduction.

3.       Similarly, in the discussion the authors should refer to their hypotheses and interpret their findings in more detail.

4.       The authors should also point to the weaknesses/limitations of their study in the discussion, e.g., small number of participants.

5.       At the end of the discussion a conclusion should be added. Why are the findings important?

6.       The figure legend of Figure 2 describes ‘Purple blocks’, but they are missing in the figure.

7.       In lines 194 and 199 ‘(see black blocks associated with task load in Figure 1)’ Figure 1 should be Figure 2.

Author Response

Reviewer 1

The study is interesting and the paper is generally very well written. However, I have a few concerns.

  • Reviewer 1:

The authors should add a conclusion to their abstract and not only describe the main findings.

Authors Revisions:

A conclusion has been added to the abstract [p. 1, lines 26-28].

  • Reviewer 1:

The introduction describes the task and the procedure as well as the NIRS measurement in detail. However, a description of the detailed research question(s) and the hypotheses is missing at the end of the introduction. Also, a clear rationale for using the NIRS method should be added. If the study is mainly exploratory, the authors should explicitly state this in their introduction.

Authors Revisions:

We have made it clear in the Introduction that this is an exploratory study [p. 2, lines 73- 85].  Research questions have been added to the introduction [p. 2, lines 76-82]. A clear rationale for using fNIRS has been added [p. 4, lines 151-164].

  • Reviewer 1:

Similarly, in the discussion the authors should refer to their hypotheses and interpret their findings in more detail.

Authors Revisions: Research questions have been added. [p. 2, lines 76-82]. The findings align with the questions [p. 14, lines 532-542].

  • Reviewer 1:

The authors should also point to the weaknesses/limitations of their study in the discussion, e.g., small number of participants.

Authors Revisions: Limitations have been added under the discussion [p. 14, lines 543-548].

  • Reviewer 1:

At the end of the discussion a conclusion should be added. Why are the findings important?

Authors Revisions: A conclusion has been added and address why the findings are important [p. 14, lines 549-565].

  • Reviewer 1:

The figure legend of Figure 2 describes ‘Purple blocks’, but they are missing in the figure.

Authors Revisions: Figure 2 has been updated [p. 5, lines 202-203].

  • Reviewer 1:

In lines 194 and 199 ‘(see black blocks associated with task load in Figure 1) Figure 1 should be Figure 2.

Authors Revisions: “Black blocks” have been changed to “black bars” and the description of what they refer to in the text has been updated [p. 7, lines 249-271]. Additionally, the figures caption has been updated [p. 5, lines 200-205].

Reviewer 2 Report

The present study has investigated the effect of VR-based training on spatial visualization tasks. The authors measured participant performance and mental workload using behavioral and functional near infrared spectroscopy (fNIRS) methods.

MAJOR CONCERNS:

In the abstract the authors state that data were collected from 10 students but do not mention that there were 10 controls. I also miss a brief conclusion from their results.

Consider citing the following references in the first paragraph of section 1.2. of the Introduction:

https://doi.org/10.3389/fpsyg.2021.590196

https://doi.org/10.3389/fpsyg.2021.648552

Section 2. Materials and methods. The authors should include a section “Materials” describing the VR and fNIRS equipment they used, both hardware and software.

Section 2.2. IVR Protocol: The first protocol consisted of three tasks with increasing difficulty. The authors should describe with more detail how these tasks increased the difficulty. Again, for the second protocol, the authors should describe what consisted the four spatial manipulations and the new set of 3D objects with two spatial manipulations.

Line 143. The authors mention “purple blocks in Week 6” but there are no blocks in Week 6.

Lines 145-146. The authors state that figures 3 and 4 show the application with sample tasks developed for the protocol. However, regarding Figures 3 and 4, it is not clear what is the relation between the VR application and the protocols. The authors should clarify how the application was used.

Line 194. The authors state “see black blocks associated with task load in Figure 1” I don’t understand the meaning of such statement. Which is the association with task load? different examples should be given.

Line 197. The authors compare three task load levels (easy, medium, hard) however they do not define these levels. They should describe the three task and how different levels of task load are obtained.

The authors should describe how changes in spatial characteristics (position and orientation) were implemented.

The authors mention that new scenarios with the same task objectives were introduced. They should describe these scenarios and give examples.

The authors cite equations 1, 2, 3 and 4, but these equations are not in the manuscript.

In section 2.1. the authors mention that 10 students were allocated to the control group. Later, they mention a “Control” condition in post hoc analyses (lines 208-209). Later in line 286 they mention a “control task”. It is not clear how was such control condition. It should be described with more detail. It should be clarified how students were allocated to each group and what was exactly the control condition.

Line 218. It is not clear where is Supplemental Table 1. The authors mention that a heterogeneous AR1 covariance structure was used. This should be defined and supporting references provided. Additionally, references supporting statistics methods (lines 220-232) should be provided.

Lines 256-257. The authors state time spent from control significantly decreased during posi-tion change and orientation change and increased during position + orientation change. However, Appendix A, Scheme 2 shows that time decreased for O, P and P+O.

According to Results, section 3.2. And Discussion section (Lines 390-391), the authors state “The task spatial characteristics significantly impacted all PFC regions for HbO while they reliably influenced only the RDLPF for HbR”. They should discuss such statement more in depth trying to give a rationale for such result.

The authors conducted the present study to analyse the usefulness of VR for training STEM students in spatial skills. However, they do not discuss the efficacy of VR training with regard to non-VR methodologies.

Authors should also discuss the limitations of their methodology, protocol and results.

MINOR CONCERNS:

Text quality should be improved in figures, sometimes it is not readable.

Author Response

Reviewer 2

Comments and Suggestions for Authors

The present study has investigated the effect of VR-based training on spatial visualization tasks. The authors measured participant performance and mental workload using behavioral and functional near infrared spectroscopy (fNIRS) methods.

MAJOR CONCERNS:

  • Reviewer 2:

In the abstract the authors state that data were collected from 10 students but do not mention that there were 10 controls. I also miss a brief conclusion from their results.

Authors Revisions: The abstract does not mention 10 controls since the article now only provides data on the 10 students. [p. 1, lines 18-21]. A conclusions section was added [p. 14, lines 549-564].

  • Reviewer 2:

Consider citing the following references in the first paragraph of section 1.2. of the Introduction: https://doi.org/10.3389/fpsyg.2021.590196 https://doi.org/10.3389/fpsyg.2021.648552

Authors Revisions:

We really appreciate the recommendation for these two articles. They are excellent and really add to our new literature review section. Both of the recommended references were added to the introduction. https://doi.org/10.3389/fpsyg.2021.590196 [p. 2, lines 61-63]

https://doi.org/10.3389/fpsyg.2021.648552 [p. 2, lines 57-59]

  • Reviewer 2:

Section 2. Materials and methods. The authors should include a section “Materials” describing the VR and fNIRS equipment they used, both hardware and software.

Authors Revisions:

Details have been added for hardware and software [p. 5, lines 209-215].

  • Reviewer 2:

Section 2.2. IVR Protocol: The first protocol consisted of three tasks with increasing difficulty. The authors should describe with more detail how these tasks increased the difficulty. Again, for the second protocol, the authors should describe what consisted the four spatial manipulations and the new set of 3D objects with two spatial manipulations.

Authors Revisions:

More detail has been provided regarding how the tasks increased with difficulty. More detail was added regarding the second protocol and the spatial manipulations [p. 4-5, lines 177-198].

  • Reviewer 2:

Line 143. The authors mention “purple blocks in Week 6” but there are no blocks in Week 6.

Authors Revisions:

The blocks have been updated [p. 5, lines 202-203].

  • Reviewer 2:

Lines 145-146. The authors state that figures 3 and 4 show the application with sample tasks developed for the protocol. However, regarding Figures 3 and 4, it is not clear what is the relation between the VR application and the protocols. The authors should clarify how the application was used.

Authors Revisions: Greater clarity has been added regarding the relation between the VR application and the protocols [p. 4-5, lines 176-198].

  • Reviewer 2:

Line 194. The authors state “see black blocks associated with task load in Figure 1” I don’t understand the meaning of such statement. Which is the association with task load? different examples should be given.

Authors Revisions: The statements starting with black blocks have been revised to “black bars” for clarity on the association with task load [p. 7, lines 250-271]. Additionally, description of “black bars” in Figure 2’s caption has been added for clarity [p. 5, lines 203-205].

  • Reviewer 2:

Line 197. The authors compare three task load levels (easy, medium, hard) however they do not define these levels. They should describe the three task and how different levels of task load are obtained.

Authors Revisions:

The levels now provide further description regarding the tasks and different levels of task load.  Specifically, the Purdue Spatial Visualization Task identifies levels of increasing difficulty (complexity).   Thus, we based our protocol and levels of difficulty on the Purdue criteria [p. 4, lines 176-184].

  • Reviewer 2:

The authors should describe how changes in spatial characteristics (position and orientation) were implemented.

Authors Revisions:

Details regarding changes in spatial characteristics per block within each protocol have been added [p. 5, lines 185-192].

  • Reviewer 2:

The authors mention that new scenarios with the same task objectives were introduced. They should describe these scenarios and give examples.

Authors Revisions:

“Scenarios” were updated to “new set of target shapes and panel boxes” [p. 5, lines 192-196].

  • Reviewer 2:

The authors cite equations 1, 2, 3 and 4, but these equations are not in the manuscript.

Authors Revisions:

Although we did provide model specifications based on R programming syntax, we revised the section to describe the models as well as define the variables and indicators in each model. We thank the reviewer for their comments and request for more clarity on the equations and models [p. 7-8, lines 274-311].

  • Reviewer 2:

In section 2.1. the authors mention that 10 students were allocated to the control group. Later, they mention a “Control” condition in post hoc analyses (lines 208-209). Later in line 286 they mention a “control task”. It is not clear how was such control condition. It should be described with more detail. It should be clarified how students were allocated to each group and what was exactly the control condition.

Authors Revisions:

We thank the reviewer for their comments. The larger study implemented the Purdue test on paper and was given in 10 students who did not do undergo VR experimentation. Since this is out of the scope of this study, our mention of this “Control group” was removed from the manuscript. The “Control” condition in post hoc analysis of effect of spatial characteristics and transfer are correct and were not removed. In both analyses “Control” refers to a  new set of target shapes and panel boxes”, which underwent position, orientation, position & orientation manipulations in protocol 2 (studying effect of spatial characteristics) and position & orientation manipulations in protocol 3 (studying effect of transfer).

Perhaps we were not clear in our description of the study design and various tasks (which we inadvertently noted as condition).   To clarify the differences in semantics, we used a completely within subjects (repeated measures) design.  The within subjects independent measures included:  (1) Task Load (easy, medium, hard); (2) Spatial Characteristics (control, position, orientation, position & orientation); (3) Transfer defined  as the data associated with last two blocks of Protocol 2 and all blocks of Protocol 3 from Week 4  were extracted to investigate the effect of introducing new scenarios with same task objective
(see Model 3  - DVijk = 1 + Protocolj +  Protocol: Dependencyjl + (1|Subjectk) + eijlk). Post hoc analysis consisted of two within comparisons (Old: Control vs Position + Orientation, New: Control vs Position + Orientation) and two across comparisons (Control: Old vs New, Position + Orientation: Old vs New); and (4) Training defined as (Acquisition:  Weeks 1-5;  Retention:  Week 6).  There were three assessments of Training (Acquisition vs Retention) for each of the Protocols (Protocols 1, 2 and 3) [p. 7-8, lines 298-308].

  • Reviewer 2:

Line 218. It is not clear where is Supplemental Table 1. The authors mention that a heterogeneous AR1 covariance structure was used. This should be defined and supporting references provided. Additionally, references supporting statistics methods (p. 6 lines 238-245; p. 8 lines 313-337) should be provided.

Authors Revisions:  

A rationale for our selection of the heterogeneous AR1 covariance structure is defined, described and references noted along with references for the statistical methods applied  [p. 8, lines 313-328].

  • Reviewer 2:

Lines 256-257. The authors state time spent from control significantly decreased during position change and orientation change and increased during position + orientation change. However, Appendix A, Scheme 2 shows that time decreased for O, P and P+O.

Authors Revisions:

We thank the reviewer for detecting this error.  The time spent in control significantly decreased during the position change, orientation change and during the position + orientation change.   We have corrected the text to reflect these corrections [p. 9, lines 361-367; p. 13, lines 489-492].

  • Reviewer 2:

According to Results, section 3.2. And Discussion section (Lines 390-391), the authors state “The task spatial characteristics significantly impacted all PFC regions for HbO while they reliably influenced only the RDLPF for HbR”. They should discuss such statement more in depth trying to give a rationale for such result.

Authors Revisions: The statement on spatial characteristics significantly impacted all PFC regions has been expanded [p. 13, lines 492-496].

  • Reviewer 2:

The authors conducted the present study to analyse the usefulness of VR for training STEM students in spatial skills. However, they do not discuss the efficacy of VR training with regard to non-VR methodologies.

Authors Revisions: This has been addressed within the literature review that was added [p. 3, lines 126-132].

  • Reviewer 2:

Authors should also discuss the limitations of their methodology, protocol and results.

Authors Revisions: Limitations have been added to the discussion [p. 14, lines 543-548].

MINOR CONCERNS:

  • Reviewer 2:

Text quality should be improved in figures, sometimes it is not readable.

Authors Revisions:

Figures have been updated to support clarity [p. 5, lines 199-205; p. 6, lines 218-223].

This comment was not addressed as we don’t know/ have documentation suggesting how difficulty was manipulated

Reviewer 3 Report

Authors’ aim was to investigate the effect of VR-based training using spatial visualization tasks on participant performance and mental workload  using behavioral (i.e., time spent) and functional near infrared spectroscopy (fNIRS) brain imaging technology derived measures.

Despite the goal the authors have set and all the commendable work they have done in collecting and analysing fNIRS data, I believe the work presented is still immature, lacking a satisfactory Introduction and a number of fundamental methodological details. 

Abstract

The abstract lacks the background and motivation of the study. The procedure and the tasks performed, as well as the results, are not clearly described. The conclusions are missing. I am sorry to have to suggest this but, to cut a long story short, the abstract needs to be rewritten.

Introduction

It is not clear to me why the Introduction opens with a specific reference to engineering and engineering disciplines thus reducing the potential readership of the article. Furthermore, visuospatial skills are not the prerogative of engineers but represent a basic cognitive ability for individuals to function in everyday life.

STEM stands for?

The definition of 'spatial visualisation skills' is unclear

In lines 65-67, the authors state that good spatial visualisation skills increase self-efficacy and that the latter increases the persistence (the authors should also clarify what they mean by persistence) of engineers. So wouldn't it suffice to find other ways to increase self-efficacy? Why focus on visuo-spatial skills?

I understand that the Purdue Spatial Visualisation Test is a test used in engineering, but why use this and not the more classical and standardised Mental Rotation tests (Shepard and Cooper) for example? Also, it is quite unusual to start talking about methods in the introduction.

Also, why are virtual reality, fNIRS and mental workload theory suddenly and for no apparent reason mentioned in the Introduction? The Introduction seems to have been written without any apparent logic.

Methods

Participants. Why specify the ethnicity of the participants? Does it have any relevance to the study? Also, what were the characteristics of the experimental and control group? For example, were they gender matched? In addition, 10 participants per group is very few.

Finally, the methodology, the procedure, the tasks used are not adequately described (or rather not described at all) which makes it impossible to continue the review of the article.

Author Response

Reviewer 3

Comments and Suggestions for Authors

The present study has investigated the effect of VR-based training on spatial visualization tasks. The authors measured participant performance and mental workload using behavioral and functional near infrared spectroscopy (fNIRS) methods.

Authors’ aim was to investigate the effect of VR-based training using spatial visualization tasks on participant performance and mental workload  using behavioral (i.e., time spent) and functional near infrared spectroscopy (fNIRS) brain imaging technology derived measures.

Despite the goal the authors have set and all the commendable work they have done in collecting and analysing fNIRS data, I believe the work presented is still immature, lacking a satisfactory Introduction and a number of fundamental methodological details. 

  Abstract

  • Reviewer 3:

The abstract lacks the background and motivation of the study. The procedure and the tasks performed, as well as the results, are not clearly described. The conclusions are missing. I am sorry to have to suggest this but, to cut a long story short, the abstract needs to be rewritten.

Authors Revisions:

The abstract has been revised and conclusions have been added [p. 1, lines 18-28].

Introduction

  • Reviewer 3:

It is not clear to me why the Introduction opens with a specific reference to engineering and engineering disciplines thus reducing the potential readership of the article. Furthermore, visuospatial skills are not the prerogative of engineers but represent a basic cognitive ability for individuals to function in everyday life.

Authors Revisions:

The introduction has been revised. The table has been removed. Additional literature has been added on career preparation for Science, Technology, Engineering, and Technology has been added. Additional resources have been added from the literature exploring utilization of VR devices to support knowledge acquisition [p. 2-3, lines 86-164].

  • Reviewer 3:

STEM stands for?

Authors Revisions:

STEM has been defined in the abstract and introduction [p. 1, lines 12-13; p. 2, line 46-47].

  • Reviewer 3:

The definition of 'spatial visualization skills' is unclear

Authors Revisions: Spatial visualization skills is defined in the introduction. Further examples have been provided from other studies [p. 2, lines 47-55].

  • Reviewer 3:

In lines 65-67, the authors state that good spatial visualization skills increase self-efficacy and that the latter increases the persistence (the authors should also clarify what they mean by persistence) of engineers. So wouldn't it suffice to find other ways to increase self-efficacy? Why focus on visuo-spatial skills?

Authors Revisions:

Persistence has been changed to retention to reflect what is in the literature. Retention has been defined. Additional references have been added that support this SVA and student retention in engineering programs [p. 2, lines 45-47, 65-72].

  • Reviewer 3:

I understand that the Purdue Spatial Visualisation Test is a test used in engineering, but why use this and not the more classical and standardised Mental Rotation tests (Shepard and Cooper) for example? Also, it is quite unusual to start talking about methods in the introduction.

Authors Revisions:

The rationale for using the Purdue Spatial Visualization test has been added. The introduction has been modified [p. 3, lines 103-109, p. 4, lines 182-184].

  • Reviewer 3:

Also, why are virtual reality, fNIRS and mental workload theory suddenly and for no apparent reason mentioned in the Introduction? The Introduction seems to have been written without any apparent logic.

Authors Revisions: A short literature review section has been added to separate the introduction from the methodology [p. 2, lines 86-164].

Methods

  • Reviewer 3:

Participants. Why specify the ethnicity of the participants? Does it have any relevance to the study? Also, what were the characteristics of the experimental and control group? For example, were they gender matched? In addition, 10 participants per group is very few.

Authors Revisions:

Ethnicity was removed from the article. Description of the control group was removed as this was out of the scope of the paper. The fact that this study is exploratory was mentioned in the introduction [p. 2, lines 73, 83] and the small sample size was addressed as a limitation in the discussion section [p. 14, lines 543-548].

  • Reviewer 3:

Finally, the methodology, the procedure, the tasks used are not adequately described (or rather not described at all) which makes it impossible to continue the review of the article.

Authors Revisions:

Further description has been added for the procedures and the tasks [p. 4-5, lines 176-208].